# Changes in the Incidence of Invasive Pneumococcal Disease in Calgary, Canada, during the SARS-CoV-2 Pandemic 2020–2022

**DOI:** 10.3390/microorganisms11051333

**Published:** 2023-05-18

**Authors:** Leah J. Ricketson, James D. Kellner

**Affiliations:** 1Department of Pediatrics, University of Calgary, Calgary, AB T2N 1N4, Canada; ljricket@ucalgary.ca; 2Alberta Children’s Hospital Research Institute, Calgary, AB T3B 6A8, Canada

**Keywords:** *Streptococcus pneumoniae*, invasive pneumococcal disease, COVID-19, SARS-CoV-2, incidence, pandemic

## Abstract

We describe the impact of non-pharmaceutical interventions (NPIs) during the COVID-19 pandemic on invasive pneumococcal disease (IPD) in Calgary. IPD declined significantly worldwide during 2020 and 2021. This may be due to the reduced transmission of and decrease in circulating viruses that often co-infect with the opportunistic pneumococcus. Pneumococcus has not been shown to frequently co-infect or cause secondary infection with SARS-CoV-2. We examined and compared incidence rates in Calgary per quarter in the pre-vaccine, post-vaccine, 2020 and 2021 (pandemic) and 2022 (late pandemic) eras. We also conducted a time series analysis from 2000–2022 allowing for change in trend at introduction of vaccines and for initiation of NPIs during the COVID-19 pandemic. Incidence declined in 2020/2021 but by the end of 2022 had begun to rapidly recover to near pre-vaccine rates. This recovery may be related to the high rates of viral activity in the winter of 2022 along with childhood vaccines being delayed during the pandemic. However, a large proportion of the IPD caused in the last quarter of 2022 was serotype 4, which has caused outbreaks in the homeless population of Calgary in the past. Further surveillance will be important to understand IPD incidence trends in the post-pandemic landscape.

## 1. Introduction

On 11 March 2020 the World Health Organization declared the SARS-CoV-2 outbreak to be a pandemic [1]. As cases spread globally, the province of Alberta implemented numerous non-pharmaceutical interventions (NPIs) to reduce transmission, including school closures and restrictions on gathering and travel [2,3]. These measures led to a decline in other infectious diseases, including invasive disease caused by *Streptococcus pneumoniae* (pneumococcus) [4].

The 7-valent pneumococcal conjugate vaccine (PCV7) was introduced for routine use in childhood vaccination programs in Alberta in 2002 and was replaced by the 13-valent pneumococcal conjugate vaccine (PCV13) in 2010 [5]. The PCV13 is also used for high-risk adults, along with the 23-valent plain polysaccharide vaccine, which is routinely indicated for all persons aged 65 years and older. Pneumococcus continues to be an important pathogen in the post-vaccine era. In Calgary, we have observed a number of outbreaks of invasive pneumococcal disease (IPD) primarily in adults who are homeless or who use illicit drugs [6,7,8]. The most recent was due to serotype 4, a vaccine serotype. Furthermore, pneumococcus continues to cause death especially in elderly populations. Although IPD has declined in children in the post-vaccine era, the remaining disease may be the most severe [9].

Invasive pneumococcal disease has some normal seasonal variation with higher rates of disease in winter [10]. The purpose of this study was to explore the impact of NPIs on incidence of IPD in Calgary during a time when there were no changes in vaccination policy or products in use. We previously reported on the change in invasive pneumococcal disease in Alberta in a brief report [11]. This study extends the data through 2022 and provides a wider context within the literature. 

## 2. Materials and Methods

### 2.1. Data Collection and Laboratory Methods 

The Calgary Area *Streptococcus pneumoniae* Epidemiology Research team (CASPER) has been collecting information on IPD in the Calgary Zone of Alberta Health Services since 1998, and this collection is ongoing [12]. Our population-based surveillance is reported through a central laboratory system (DynaLIFE, formerly Calgary Laboratory Services). The laboratory system notifies CASPER of IPD cases detected by culture or molecular testing. A research nurse conducts a review of electronic health records for positive cases using a standardized case report form. The local institutional review board has provided a waiver of consent for this limited health record review. In earlier years of data collection, a patient interview was conducted, and a more extensive case report form was completed. During that phase of the study, written informed consent was obtained from all participants. Patients who do not live within the Calgary Health Zone are excluded from the study population.

Isolates are identified as pneumococcus using colony morphology, alpha hemolysis, optochin susceptibility, and bile solubility. Antimicrobial susceptibility and serotyping are performed on all culture-positive samples. Serotyping is conducted using the Quellung reaction with serotype-specific anti-sera from Statens Serum Institute (Copenhagen, Denmark) [13]. Minimum inhibitory concentrations (MICs) are conducted for all viable isolates using broth microdilution following the Clinical Laboratory Standards Institute guidelines for susceptibility [14]. Isolate and patient information are collected and stored in a secure FileMaker Pro 19 database. All cases of IPD occurring in the Calgary Zone between 2000 and 2022 were included in the analysis.

### 2.2. Incidence and Incidence Rate Ratios 

Here we report quarterly incidence of IPD in Calgary during the era of the COVID-19 pandemic caused by the SARS-CoV-2 virus, starting in early 2020, compared with quarterly incidence in the years pre- and post-introduction of the conjugate vaccines. Incidence was calculated using the number of cases per 100,000 people per quarter. Calgary population estimates for each year came from the Alberta Interactive Health Data application [15]. We took the average incidence per quarter for 2000–2002 (the pre-vaccine era), 2003–2010 (PCV7 era), and 2011–2019 (PCV13 era); we reported 2020 and 2021 (SARS-CoV-2 pandemic) and 2022 (late pandemic/re-opening) separately, and we graphed them to visualize how the average incidence per quarter during and after the pandemic compared to past eras. We also stratified by age groups (<18 years and ≥18 years). We reported crude incidence rate ratios (IRRs) using the “iri” command in Stata to compare incidence between key periods to show significant differences between values. We compared each quarter of 2020, 2021, and 2022 to the corresponding quarter in 2019. We also compared the pre-PCV7 October to December quarter to the corresponding quarter of 2022. IRRs are reported with 95% confidence intervals (95% CI), and *p*-values below an alpha of 0.05 were considered significant. 

### 2.3. Interrupted Time Series Analysis

We conducted an interrupted time series analysis looking at all ages’ quarterly incidence of IPD in Calgary over time from 2000–2022. The benefit of a time series analysis is it allows for evaluation of a health intervention (e.g., vaccine) or the effect of an ecological change (e.g., pandemic). This is achieved by setting the model to allow for a change in the linear trend at a defined point in time. We used the calculated incidence per quarter over 2000–2022 as the data points for the time series analysis and utilized the “itsa” command in Stata/SE version 17.0. We allowed for change in trend (interruptions) at three time points: the time of vaccine introductions (PCV7 change at quarter 4, 2002, and PCV13 change allowed at quarter 4, 2010) and at the introduction of NPIs early in the COVID-19 pandemic (change allowed at quarter 2, 2020). We assessed for autocorrelation of residuals visually using the “ac” command in Stata/SE version 17.0 and checked for significance using the Cumby–Huizinga general test for autocorrelation of residuals in a time series. We then adjusted the interrupted time series for a lag of 4 according to the autocorrelation results. Interrupted time series results are reported with 95% confidence intervals (95% CI), and *p*-values below an alpha level of 0.05 were considered significant. The results show the change in level at each interruption and the change in linear trend. 

Stata/SE version 17.0 was used for calculating IRRs and conducting interrupted time series analysis. Incidence graphs were generated using Microsoft Excel version 16.72. The study protocol was approved by the Ethics Department of the University of Calgary (Ethics ID REB15-0571).

## 3. Results

### 3.1. IPD Incidence in Calgary 

Between 2000 and 2022, 2873 people with IPD presented to Calgary hospitals. Of these cases, 397 were in children and 2476 were in adults. In 2020, When NPIs were in place, Calgary IPD incidence went down to less than one case per 100,000 people per quarter where the expected incidence in the post-PCV7 and post-PCV13 eras would be between two and three cases per 100,000 people per quarter during the non-summer quarters (Figure 1). In 2021, IPD stayed low, with less than 1 case per 100,000 people per quarter, then began to increase in the last quarter of 2021 up to 2.5 cases per 100,000 people. In 2022, the incidence for the first and second quarters had returned to equivalent incidence rates to the PCV7 era. By the last quarter of 2022, incidence of IPD increased substantially in adults with five cases per 100,000 people per quarter in the October to December quarter. This incidence is higher than the incidence of the October to December quarter in the pre-vaccine era (three cases per 100,000 people per quarter) (Figure 1). 

In children under 18 years of age, there were only four cases per year in 2020, of which only two occurred after the NPIs were put in place, and five cases per year in 2021 (Figure 2a). This is in contrast to past years where we observed between 14 and 25 cases per year in children in the post-PCV era. There was a 71% decrease in cases from 2019 to 2020, and 2019 was already a relatively low year with only 14 cases reported in children. In 2022, after NPIs were removed, there were 26 cases per year in children.

#### 3.1.1. Incidence Rate Ratios Comparing Pre-Pandemic (2019) and Early Pandemic Periods (2020/2021)

The crude IRR for IPD at all ages for the first quarter of 2020 compared to the first quarter of 2019 was 1.3 (95% CI: 0.8–2.2; *p* = 0.2607). Similarly, for the second, third, and fourth quarters of 2020, the IRR compared to the corresponding quarters of 2019 were 0.2 (95% CI: 0.1–0.4; *p* < 0.001), 0.6 (95% CI: 0.2–1.3; *p* = 0.1575), and 0.6 (95% CI: 0.3–1.0; *p* = 0.0402), respectively. 

For 2021 compared to 2019, the IRR for the first quarter in all ages was 0.2 (95% CI: 0.1–0.5; *p* = <0.001), for the second quarter was 0.2 (95% CI: 0.1–0.3; *p* < 0.001), for the third quarter was 0.9 (95% CI: 0.4–2.0; *p* = 0.8061), and for the fourth quarter was 1.1 (95% CI: 0.7–1.7; *p* = 0.7496).

#### 3.1.2. Incidence Rate Ratios Comparing Pre-Pandemic (2019) and Pre-PCV7 Periods to Later Pandemic Period (2022)

For 2022 compared to 2019, the IRR for the first quarter in all ages was 0.9 (95% CI: 0.6–1.6; *p* = 0.8267), for the second quarter was 0.8 (95% CI: 0.5–1.2; *p* = 0.2487), for the third quarter was 1.8 (95% CI: 0.9–3.5; *p* = 0.0623), and for the fourth quarter was 2.1 (95% CI: 1.4–3.1; *p* < 0.001).

In the later pandemic era, the IRR between the average incidence of the October to December quarter of the pre-PCV7 era and the October to December quarter of 2022 in the later pandemic period was 1.5 (95% CI: 0.99–2.4; *p* = 0.0420).

### 3.2. 2022 Serotypes Causing IPD in Calgary 

The resurgence of IPD in 2022 was caused by a variety of serotypes, though serotype 4 was the most common cause of infection by far, causing 49/172 (28.4%) cases in adults; however, serotype 4 caused 0 cases in children (Table 1). Serotype 3 was the next most common with 22/172 (12.8%) cases in 2022. Serotypes 7F, 9N, and 20 each accounted for 12 (6.9%), 11 (6.4%), and 10 (5.8%) cases, respectively. In 2019, serotype 4 accounted for only four cases, while serotype 7F (15.3%) and 3 (13.7%) accounted for the highest proportion of cases the year before the pandemic and NPIs. 

### 3.3. Interrupted Time Series Analysis 2000–2022

The interrupted time series analysis allowed for change in incidence rate at the introduction of PCV7, at the introduction of PCV13, and at the time of initiation of NPIs in March 2020. In the post-PCV7 era, overall incidence had a decreasing trend but was non-significant (−0.03 95% CI: −0.09–0.02, *p*-value = 0.2334) (Figure 3). In the post-PCV13 era there was a significant increase in rate of IPD of 0.02 (95% CI: 0.005–0.03) per 100,000 per quarter (*p* < 0.001). In the April–June quarter of 2020, after initiation of NPIs, there was a significant drop in the level of the incidence rate of IPD per quarter by 2.4 (95% CI: 1.6–3.3) cases per 100,000 per quarter (*p* < 0.001). The incidence rate trend then increased again over the course of the pandemic with a quarterly increase of 0.3 (95% CI: 0.2–0.4) cases per 100,000 per quarter (*p* < 0.001).

## 4. Discussion

The public health efforts to mitigate the spread of SARS-CoV-2 also resulted in historically low rates of pneumococcus, influenza, and other respiratory viruses and bacteria in multiple countries [4,16,17], including Canada [18,19]. A large prospective surveillance study showed a decline in *S. pneumoniae*, *Haemophilus influenzae*, and *Neisseria meningitidis* in 26 countries during the time when public health containment efforts were in place to decrease the spread of SARS-CoV-2 [4]. Smaller studies reported decreases in IPD, community-acquired pneumonia, and bloodstream infections due to pneumococcus in multiple countries [20,21,22,23,24,25,26,27,28,29,30].

In Calgary, January to March of 2020 began with normal post-PCV incidence rates of IPD, followed by a sharp decline in incidence for quarters 2 and 3 in April to September, as evidenced by the very low IRR comparing 2020 with the previous year, and much lower than the usual seasonal decline in the third quarter of 2020. This is further shown in interrupted time series analysis as a significant drop in cases at the interruption allowed for implementation of NPIs. The incidence of IPD increased in the fourth quarter of 2020, but not to normal rates. The incidence in children, which was already much lower in the vaccine era compared to the pre-vaccine era, declined by 71% in 2020 compared to 2019 and was low for the entire year, with a total of four cases in 2020, of which only two cases occurred after NPIs were first implemented in March 2020. The decline in adults was only apparent from the second quarter onwards. This change in incidence occurred within the context of continual routine surveillance for pneumococcal disease and normal vaccine uptake within the community. Prior to the pandemic, we observe with the interrupted time series analysis a slow increase in IPD cases per 100,000 per quarter in the post-PCV13 period, followed by a sudden drop in cases at initiation of NPIs and then a stronger trend of increasing incidence of IPD per 100,000 cases per quarter as the pandemic resolved and NPIs were removed. Considering we were observing a slight increasing trend in incidence prior to the pandemic, this is further evidence that the NPIs were the cause of the sudden drop in incidence.

In 2022, during the first and second quarter, rates returned to normal and were comparable to 2019, as shown by no significant difference in IRR between these quarters. However, in the second and third quarter of 2022, rates of IPD increased to above what would be expected and were higher than 2019 with an IRR of 2.1 when comparing 2019 and 2022 incidence for the October to December quarter.

Initially it was assumed that the decline in IPD was a result of decreased transmission of pneumococcus within communities [21,23]. However, Weinberger et al. examined seasonality of IPD to determine whether it was associated with changes in viral activity or with seasonal increases in pneumococcal carriage [10]. Their findings suggested that seasonal variation in non-pneumonia IPD was associated with changes in carriage prevalence [10]. However, pneumococcal bacteremic pneumonia was associated with increased viral activity in the community [10]. Weinberger et al. suggested that this means carriage alone is not sufficient to lead to bacteremic pneumonia; a viral illness or other seasonal stressor is also involved [10]. Bacteremic pneumonia is by far the most common manifestation of IPD and so reduction of this would create a reduction in overall IPD even if levels of other IPD manifestations did not decline. This may provide insight into why IPD declined substantially during the period of public health restrictions during the COVID-19 pandemic. The NPIs implemented to control SARS-CoV-2 transmission also reduced transmission of other respiratory viruses worldwide that cause infections and may lead to secondary infections caused by pneumococcus [17]. A Canadian surveillance study showed a reduction of all viruses following implementation of NPIs in Canada [19].

The importance of antecedent viral infections on the incidence of IPD is further supported by a prospective cohort study by Danino et al. in Israel. They described how invasive disease was reduced during the implementation of NPIs to control COVID-19, but nasopharyngeal carriage prevalence stayed the same in children <5 years of age [31]. This suggests that it was the reduction in the prevalence and transmission of viruses such as RSV, influenza, and human metapneumoviruses, which are frequently involved in co-infections with pneumococcus, that caused the decrease in IPD rather than reduced transmission of pneumococcus [31]. Viruses are understood to affect host immunity leading to opportunistic infection by bacteria that are normally carried in the respiratory tract as commensals [32,33,34]. Another study in France made similar observations, showing a 63% reduction in IPD caused by non-PCV13 serotypes with high disease potential and a 53% reduction in non-PCV13 serotypes with low disease potential without a significant change in pneumococcal carriage rates (−12%, 95% CI: −37–12%, *p* = 0.32) [35].

Another factor that may have contributed was a reduction in presentation to hospital for those with less severe IPD, as many people avoided hospitals during the beginning of the SARS-CoV-2 pandemic [36]. An analysis in the Netherlands suggested it went beyond just transmission, and the decline in observed IPD was affected by collection and capture of IPD cases as well [37]. This could have been true in Calgary for milder cases as all the current surveillance for IPD is conducted through hospital and clinic visits.

The association of IPD with SARS-CoV-2 infections is uncommon [20,38,39,40,41,42,43]. There are an increasing number of reports describing this lack of association [20,42,43,44]. In a systematic review of co-infection by SARS-CoV-2 and other respiratory pathogens, the results showed influenza A to be the most common pathogen to appear in co-infection with SARS-CoV-2 [38], and a review by Russell et al. described *Staphylococcus aureus* and *Haemophilus influenzae* as the most common bacteria involved in co-infections with SARS-CoV-2 [42]. However, one study suggested that when co-infection with pneumococcus did occur, odds of death were over 7-fold higher, particularly in older adults [20]. Similarly, case reports with co-infection by pneumococcus and SARS-CoV-2 reported very severe infection [40,45]. The lack of association of pneumococcus causing co-infection with SARS-CoV-2 may further explain the reduction of IPD during the pandemic where SARS-CoV-2 was the primary virus circulating worldwide. A study in Barcelona described that the percentage of patients with co-infection on admission was low (3.1%), but *S. pneumoniae* and *S. aureus* were the most common pathogens [46]. A review by Westblade et al. reported that although bacterial co-infections were uncommon on admission with SARS-CoV-2 (<4%), they more frequently occurred in patients with prolonged hospitalizations or ICU stays (6–29%) [47].

The observed ecological variation in IPD incidence during the COVID-19 pandemic is important in the pneumococcal conjugate vaccine era. The United Kingdom modified their vaccine schedule to 1 + 1 doses at the start of 2020; therefore, data will need to be cautiously interpreted from years where vaccine schedule change was implemented during the pandemic [48]. The dramatic decline in IPD incidence in our study in Calgary, where vaccination policy and availability had not changed, suggests that the incidence of IPD in 2020 and 2021 in particular could not be attributed to any change in vaccine schedules.

Of interest in our study was our observation of the rapid recovery of IPD incidence in 2022. IPD incidence in adults recovered to levels of the pre-PCV13 era and, by the October to December quarter of that year, incidence rates in adults exceeded levels of the pre-vaccine era. Furthermore, interrupted time series analysis showed a small but significant increasing trend in incidence prior to the drop in incidence associated with the pandemic and NPIs, and a stronger increasing trend after the NPIs were removed. This alarming recovery of IPD will require continued surveillance to understand whether Calgary is experiencing another outbreak like we have observed in the past with serotypes 5 and 4 [7,8], or whether the post-COVID environment has somehow brought a new normal for IPD incidence. Other jurisdictions have also observed an increase in IPD above the post-vaccine baseline in the late pandemic era [49,50]. This too may suggest that the fluctuations in IPD are related more to the changes in viral activity than to the changes in public health behaviours and transmission of pneumococcus. The winter of 2022 had abnormally high incidence of viral activity following the removal of NPIs [51,52]. This increase in pneumococcal disease may be partly related to delays in childhood vaccinations due to the pandemic [53,54]. However, we found the rates in children, the population who receive the vaccine, remained fairly low or to where we would expect in the post-vaccine era, but rates in adults increased to above pre-vaccine levels. We also observed serotype 4, a vaccine serotype, to be the most prevalent serotype in adults in 2022 (28.4%). Serotype 4 was involved in a recent outbreak in Calgary in 2018 that was largely in adults living homeless or using illicit drugs [7]. Serotype 3 and serotype 7F, also vaccine serotypes, continue to persist in adults in Calgary despite widespread childhood vaccination.

We were limited in this analysis in that we used the population of Calgary as our denominator, which assumes the entire population to be at risk. This may not be true, as carriage rates would affect the number of people at risk of IPD. However, this is true for all years, and so the relative incidence is the same throughout and therefore can show differences between years and comparative changes throughout the pandemic. We also have low numbers of pneumococcal disease within the Calgary region each year; therefore, we did not have the numbers to look more closely at children. Our incidence rate ratios were a simple comparison of crude incidence rates, and we did not account for confounding factors.

## 5. Conclusions

IPD declined in Calgary and worldwide as a result of NPIs and the effect of diminished transmission of pneumococcus combined with the decline of viruses that frequently co-infect with pneumococcus. The rapid increase in IPD following the re-opening of Calgary schools and businesses indicates the importance of continued monitoring to understand the trends in IPD and distinguish outbreaks and seasonal fluctuations from true changes in baseline incidence. This is especially important with the continuing changes in vaccine availability and acceptance and with new higher-valent pneumococcal conjugate vaccines on the horizon [55].

## Figures and Tables

**Figure 1 microorganisms-11-01333-f001:**
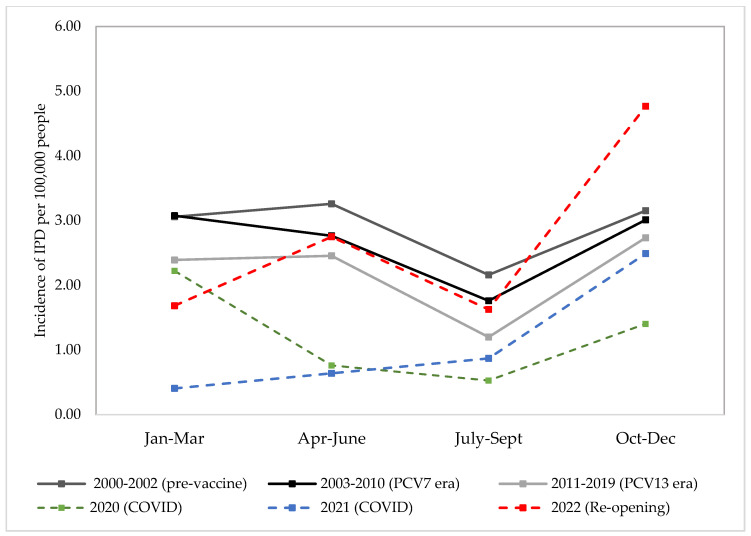
Average crude incidence per quarter of IPD in all ages for pre-vaccine era (2000–2002), PCV7 era (2003–2010), PCV13 era (2011–2019), SARS-CoV-2 pandemic (2020 and 2021), and late pandemic (2022).

**Figure 2 microorganisms-11-01333-f002:**
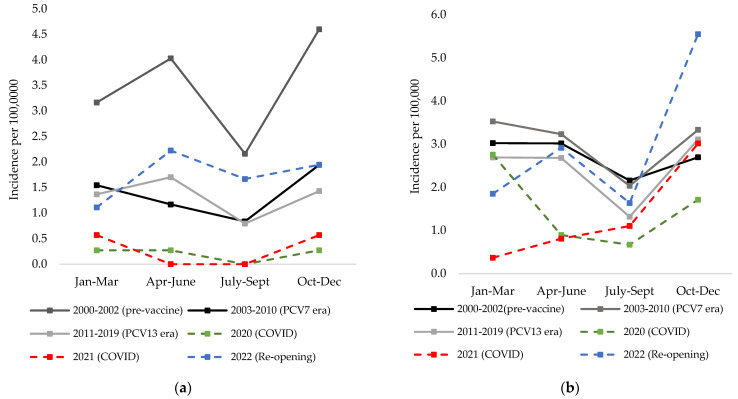
Average crude incidence per quarter of IPD for pre-vaccine (2000–2002), PCV7 era (2003–2010), PCV13 era (2011–2019), SARS-CoV-2 pandemic (2020 and 2021), and late pandemic (2022): (**a**) in children less than 18 years of age, and (**b**) in adults 18 years and older.

**Figure 3 microorganisms-11-01333-f003:**
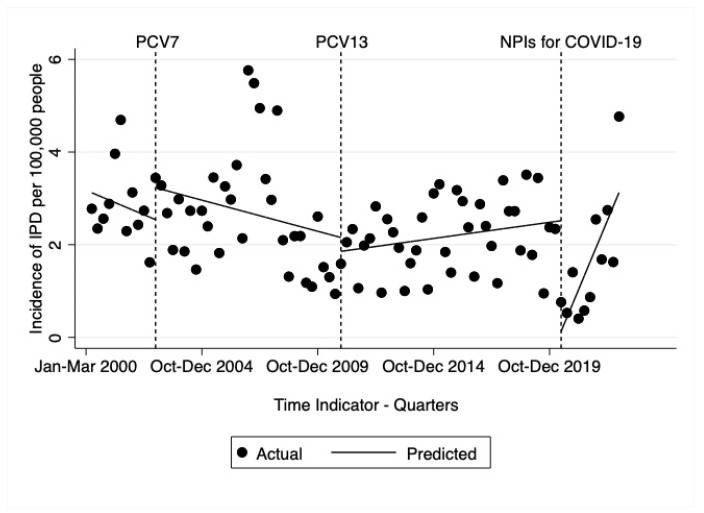
Interrupted time series analysis with interruptions for vaccine introductions (PCV7 and PCV13) and initiation of non-pharmaceutical interventions due to the SARS-CoV-2 pandemic.

**Table 1 microorganisms-11-01333-t001:** Serotypes causing IPD in 2019–2022 stratified by year and age (children <18 years and adults ≥18 years of age) and grouped by vaccine serotypes.

	2019		2020		2021		2022	
Serotype	Children	Adults	Children	Adults	Children	Adults	Children	Adults
7-valent vaccine serotypes								
4	0	4	0	5	0	11	0	49
6B	0	0	0	0	0	0	0	0
9V	0	0	0	1	0	2	0	7
14	0	0	0	0	0	0	0	1
18C	0	0	0	0	0	0	0	0
19F	1	2	1	1	0	1	3	1
23F	0	0	0	0	0	0	0	0
10-valent vaccine serotypes								
1	0	0	0	0	0	0	0	0
5	0	0	0	0	0	0	0	0
7F	0	20	0	10	1	7	0	12
13-valent vaccine serotypes								
3	2	18	0	8	0	8	2	22
6A	0	1	0	0	0	0	0	0
19A	0	2	0	1	0	0	0	7
15-valent vaccine serotypes								
22F	4	12	1	6	1	6	2	8
33F	0	3	0	0	0	1	0	3
20-valent vaccine serotypes								
8	1	2	0	2	0	4	1	5
10A	0	1	1	0	0	0	0	4
11A	0	2	0	3	0	0	2	4
12F	0	6	0	9	0	0	0	2
15B	0	3	0	0	1	0	2	1
23-valent polysaccharide vaccine serotypes								
2	0	0	0	0	0	0	0	0
9N	1	8	0	12	0	9	0	11
17F	0	1	0	0	0	0	1	0
20	0	11	0	4	0	4	2	10
Other serotypes								
6C	0	2	0	1	0	3	0	1
7C	0	1	0	1	0	1	0	1
13	0	2	0	0	0	1	0	1
15A	0	4	0	2	0	2	0	2
15C	0	1	0	1	0	0	1	1
16F	0	3	0	2	0	1	0	2
21	0	0	0	0	0	0	1	0
23A	2	4	0	4	0	2	0	0
23B	3	2	0	0	0	2	0	4
28A	0	0	0	1	0	0	0	1
29	0	0	0	0	0	0	1	0
31	0	7	0	3	1	3	0	1
34	0	1	0	0	0	0	0	1
35B	0	4	1	1	0	1	2	0
35F	0	1	0	0	0	0	1	1
38	0	3	0	0	0	0	0	0
Missing	0	0	0	0	1	1	5	9
Total	14	131	4	78	5	70	26	172

## Data Availability

The data presented in this study are available on request from the corresponding author. The human research participant data are not publicly available due to ethical restrictions.

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
