# Peer review of "Changes in the Incidence of Invasive Pneumococcal Disease in Calgary, Canada, during the SARS-CoV-2 Pandemic 2020–2022"

_microorganisms, 2023, doi:10.3390/microorganisms11051333_

Round 1
Reviewer 1 Report
The article shows changes in the incidence of IPD in Calgary, Canada, during the the SARS CoV 2 pandemic. CASPER data show a decrease in IPD during 2020 and 2021 in children and adults. I consider it an interesting study, which contributes to the knowledge of the epidemiology of IPD and the impact that the pandemic had on other respiratory diseases.

Author Response
Thank you for your positive review and support of this manuscript. We can appreciate the challenge of reading Table 1 and have attempted to make it simpler by removing 0s as you suggested.
Reviewer 2 Report
I enjoyed reading this wor. The authors gave a detailed insight of the decline of IPD in Calgary during SARS-Covid-19.
I suggest the authors go over the text and correct minor mistakes.
The authors report on the changes of invasive pneumococcal disease in Calgary, Canada during SARS-Cov-2 pandemic. The work presented looks at post vaccine, during the pandemic and post pandemic. The authors also conducted a time series analysis (2000-2022) and looked at the trend at vaccine introduction and initiation of non-pharmaceutical intervention during the pandemic. The authors observed large proportion of the invasive pneumococcal disease was reported were caused by Serotype 4. The study is aimed for public health.
2. Study was comprehensive; however, it is only specific to one region of Canada, Calgary. It would have been appealing to the wider readers if the study were to report data showing changes in IPD incidence of the whole of Canada. However, I still believe this would be appealing to Calgary public health and give insights on what serotype to monitor in the near future.
3. Minor correction:
I have noticed minor text that needs corrects such as:
1. all the statistic need to have an italic “P”.
2. I have noticed some words need spacing
3. The authors need to include a section on how statistics were carried out
I found minor errors in the text, authors should correct this
Author Response
Thank you for your supportive and constructive review. We have made an effort to address your comments.
- We agree that it would be ideal to have a larger area of Canada covered, however the Calgary-Area Streptococcus pneumoniae epidemiology research program only collects data within the Calgary Zone of Alberta Health Services so across Canada is not available to us. We feel this still adds to the existing literature by showing that changes in IPD were common across many cities and countries during the Sars-CoV-2 pandemic, while also highlighting the regional differences that can occur.
- Minor text edits: We have attempted to rectify the text according to your comments. We italicized for p-values, but we are not entirely clear on what your concerns were around word spacing. We have checked to ensure spacing is appropriate to Microorganisms formatting and hope we have remedied any issues.
- We do have a section on how statistics were conducted. We have added to the section and hope this corrects your concerns. We also created subsections for the analysis and hope this gives more clarity.